# Placenta Accreta Spectrum (PAS) Disorder: Ultrasound versus Magnetic Resonance Imaging

**DOI:** 10.3390/diagnostics12112769

**Published:** 2022-11-12

**Authors:** Ida Faralli, Valentina Del Negro, Alessandra Chinè, Natalia Aleksa, Enrico Ciminello, Maria Grazia Piccioni

**Affiliations:** 1Department of Maternal and Child Health and Urological Sciences, “Sapienza” University of Rome, 00161 Rome, Italy; 2Department of Statistical Sciences, “Sapienza” University of Rome, 00185 Rome, Italy

**Keywords:** placenta accreta spectrum, diagnostic ultrasound, placental pathology, prenatal diagnosis, magnetic resonance

## Abstract

Objective: PAS is one of the most dangerous conditions associated with pregnancy and remains undiagnosed before delivery in from half to two-thirds of cases. Correct prenatal diagnosis is essential to reduce the burden of maternal and fetal morbidity. The purpose of our study is to evaluate the accuracy of US and MRI in the diagnosis of PAS. Study design: In this retrospective study, 104 patients with suspected placenta accreta were enrolled and had been investigated with US and MRI. They were divided into four groups: no PAS, accreta, increta, and percreta. Results: Compared to MRI, US results were higher in the diagnosis and in the identification of PAS severity (85% US vs. 80% MRI). For both methods, in the case of posterior placenta, there is greater difficulty in identifying the presence/absence of the disease (67% in both methods) and the severity level (61% US vs. 55% MRI). Conclusion: US, properly implemented with the application of defined and standardized scores, can be superior to MRI and absolutely sufficient for the diagnosis of PAS, limiting the use of MRI to a few doubtful cases and to cases in which surgical planning is necessary.

## 1. Introduction

Placenta accreta spectrum (PAS) disorder comprehends a group of anomalies characterized by an abnormal adhesion or invasion of the trophoblastic tissue to the uterine myometrium and serosa [1].

The spectrum encompasses placenta accreta (attachment of the placenta to the myometrium without interposed decidua), placenta increta (invasion through the myometrium), and placenta percreta (invasion through serosa and eventually other adjacent structures) [2].

Since the first description of PAS, the incidence has progressively increased, paralleling the rise in cesarean section (CS) rates that represents the main risk factor [3,4]. PAS is now estimated to occur in 3 out of 1000 pregnancies, showing a 5-times incidence increase since the 1970s [5]. Concerning prevalence of PAS disorders, no exact data can be reported because of the lack of epidemiological information in low-income countries [6].

PAS is one of the most dangerous conditions associated with pregnancy because it can cause massive hemorrhage leading to multisystem organ failure, disseminated intravascular coagulation, the necessity for an intensive care unit, hysterectomy, and even death. It represents the first cause for both a hysterectomy associated with a caesarean section and a peripartum hysterectomy [2]. Diagnostic difficulties are still high and PAS remains undiagnosed in between half [7,8] and two-thirds [4,9] of cases.

The necessity for pre-surgery planning in case of PAS enhances the importance of accurate prenatal diagnosis, which is essential to reduce the burden of maternal and fetal morbidity associated with this pathology [10,11].

Although FIGO classification for clinical diagnosis [12] and a consensus regarding pathologic diagnosis of PAS [13] were recently released, there is no agreement in the literature concerning the prenatal diagnosis of PAS [14]. Currently, diagnosis is primarily based on ultrasound (US), which has proved to be very reliable when performed by experienced operators. US findings suggesting PAS disorders described by grey-scale ultrasound imaging are the loss of the hypoechoic retroplacental (clear) zone because of the abnormal extensions of placental villi into decidua, abnormal placental lacunae, myometrial thinning, hyperechoic uterus–bladder interface, exophytic mass, and placental bulge. Furthermore, color Doppler findings are as follows: hypervascularization patterns between placental, placental basal plate, subplacental zone or undelying tissues; bridging vessels; and diffuse or turbolent flow in lacunae [15]. However, the added value of magnetic resonance imaging (MRI) has not yet been clearly established. MRI signs of PAS are uterine bulging, heterogeneous signal intensity, dark intraplacental bands on T2, focal interruption of myometrium, and tending of the bladder [15].

Moreover, although US and MRI have shown comparable overall diagnostic accuracy and the most recent FIGO recommendations have defined MRI as “non-essential” [15,16], diagnostic confirmation with MRI is primarily performed in tertiary-care centers [17,18,19].

The purpose of our study is to evaluate the diagnostic accuracy of US and MRI in the diagnosis of PAS and to define the most relevant characteristics that can predict placental invasion.

## 2. Materials and Methods

### 2.1. Patient Population

We retrospectively reviewed medical records of patients referred to the Department of Gynecology and Obstetrics, Policlinico Umberto I (Rome), from January 2014 to January 2020 for suspected placenta accreta. Our population included 104 pregnant women with at least one previous caesarean section and placenta previa or low-lying placenta [20] who had been investigated in the third trimester with transabdominal/transvaginal US, color Doppler examination, and MRI.

All patients enrolled signed a written informed consent.

### 2.2. Imaging Acquisition

The US equipment included an ultrasound machine with a 3.5 MHz transabdominal probe and a 7 MHz transvaginal probe (GE Voluson^®^ 730, GE Medical Systems, Zipf, Austria).

The MRI study was performed using a Siemens Magneton Avanto 1.5 T device. For accurate placental localization, acquisition was carried out using a technique called HASTE (half-Fourier acquisition with single-shot turbo spin-echo) in apnea, with sagittal, axial, and coronal scans (4 mm thickness without gaps and a receiver band width of 62.5 KHz). Localization, extension, and topography of placental invasion were evaluated.

### 2.3. Imaging Analysis

US and MRI images were reviewed by two experts with more than 5 years of experience in the diagnosis of PAS.

For US images, the diagnosis was made using an ultrasound score developed by our working group [21], previously described and based on ultrasound descriptors suggested by the European Working Group on Abnormally Invasive Placenta (EW-AIP), stratified by severity score and combined with anamnestic data (Figure 1).

For MRI images, the diagnosis was based on features previously described in the literature as useful for predicting PAS [22].

### 2.4. Definitive Diagnosis

Diagnosis of placental invasion was based on histological confirmation. In all cases in which hysterectomy was avoided, the identification of accretion was carried out during surgery by expert surgeons of our institute.

Based on histological examination and/or surgical evaluation, the patients were divided into four groups: Group 0 (absence of PAS), Group 1 (placenta accreta), Group 2 (placenta increta), and Group 3 (placenta percreta).

### 2.5. Statistical Analysis

For both US and MRI, sensitivity and specificity were calculated using tests in binary diagnostic tables. Statistical analysis was performed using the software R v. 3.6.3 (29 February 2020) “Holding the Windsock”.

The study was conducted according to the guidelines of the Declaration of Helsinki.

Ethical review and approval were waived for this study because of its retrospective design.

## 3. Results

During the 5 years considered, 104 patients who met the inclusion criteria were enrolled and subsequently distributed into four groups according to the definitive histological or surgical diagnosis (respectively, 80 and 24).

First of all, we analyzed patients’ demographic characteristics and risk factors for PAS for each group [Table 1].

To establish the impact of risk factors for PAS (number of previous CS, previous courettage, previous myomectomy, and presence of placenta previa major) on the probability of developing the disease, a logistics model was implemented [Table 2].

The presence of major placenta previa and previous myomectomy showed a statistically significant impact (*p* < 0.05) on the probability of developing PAS; anyway, previous CS was the factor with the greatest impact on the risk of developing PAS (*p* < 0.001). As the number of CS increases, the finding of PAS increases too.

For the analysis of the results, we first compared the two techniques in diagnosing absence/presence of PAS and consequently in diagnosing the degree of invasion.

Subsequently, we stratified patients according to their placental localization.

### 3.1. Absence/Presence of PAS

Both US and MRI overestimated nine no-PAS cases. Among the PAS cases, US underestimated two cases while MRI underestimated three cases [Table 3].

We then calculated accuracy, sensitivity, specificity, and positive and negative predictive values of the two methods [Table 4].

Compared to MRI, US results were higher for all the indicators considered, but not statistically significant.

### 3.2. Diagnosing Grade of PAS

We then analyzed the ability of the two methods to define the depth of invasion (Table 5). Among 25 PAS cases, US underestimated invasion degree in 5 cases (2 cases of placenta accreta and 3 cases of placenta increta). Moreover, diagnosis was overestimated in 2 cases of placenta increta diagnosed as percreta.

MRI, instead, underestimated invasion’s degree in 8 cases (2 cases of placenta accreta, 5 cases of placenta increta, and 1 case of placenta percreta) and overestimated it in 2 cases of placenta increta.

US accuracy in recognizing the severity level of disease was 0.85, while MRI accuracy was 0.80, which is not statistically significant.

So, US had the highest diagnostic accuracy in differentiating PAS severity, confirmed by the value of the chi-squared test statistic equal to 147.05 for ultrasound and 104.9 for MRI.

We also stratified patients according to their placental location because, as reported in the literature, MRI could offer valid help in the case of posterior placenta [23,24,25,26,27,28,29,30,31].

Comparison between MRI and US in PAS evaluation considering placental localization is reported in Table 6.

MRI and US ability in evaluation of PAS degree considering placental localization is reported in Table 7.

For both methods, identifying presence, absence, and severity of PAS, in the case of posterior placenta, is difficult.

While accuracy is the same in both methods, US has slightly better results in identify the level of severity (61% vs. 55%); this difference was not statistically significant according to the chi-square test (*p* = 0.1766).

So, US showed better performance than MRI both in diagnosis of PAS and in characterizing the PAS’s degree, but the *p* value was not statistically significant.

## 4. Discussion

Prenatal diagnosis of PAS is still a subject of strong debate in the literature. Currently, the diagnosis is based on US, which has a good sensitivity and specificity if performed by experienced operators. Several studies have compared ultrasound and MRI for the diagnosis of PAS [11,25,26,31,32,33,34,35]. Despite the heterogeneity of the studies and the results obtained, MRI is defined as a non-essential method in the diagnosis of PAS but useful in doubtful cases on ultrasound.

In our study, according to the literature, US shows a good performance in identifying the presence of PAS both in anterior and posterior placenta with an accuracy of 94% vs. 67% (*p* = 0.0024), respectively; MRI accuracy results are almost the same for both anterior and posterior placenta (93% vs. 67%, respectively, *p* = 0.0055). Therefore, the use of a US-standardized score offers good results in PAS/no-PAS diagnosis considering all the parameters analyzed [Table 4].

Additionally, in the evaluation of the PAS degree in our study, US is better than MRI despite the non-statistically significant *p* value. This disagrees with some of the data reported in the literature [36,37] that consider MRI to be more accurate in distinguishing the depth of placental invasion.

D’Antonio et al. in 2013 [18] analyzed 23 studies (3707 patients) with the aim of defining US performance in the diagnosis of PAS in high-risk patients. In women with low anterior placenta and previous uterine surgery, a third-trimester US is highly specific and sensitive in the diagnosis of PAS. The same research group also conducted another meta-analysis including 18 studies (1010 patients) to define the diagnostic accuracy of MRI in PAS diagnosis [38]. They stated that MRI accuracy in PAS diagnosis is comparable to US. However, MRI can define the topography of placental invasion necessary for surgical planning and should always be performed if lateral invasion is suspected on US.

Another difference with the literature concerns posterior placentation. In most of the studies, MRI is recommended for the study of patients with posterior placentation [23,24,25,26,27,28,29,30,37]. In our data, US was superior to MRI in recognizing the presence of PAS in posterior placenta, but the difference is not statistically significant.

So, our study demonstrates how US, properly implemented, can be useful in the diagnosis of PAS, limiting the use of MRI, which is more expensive and difficult, in case of the necessity of surgical planning or in doubtful cases [17]. We hope that improvement of US techniques, with the application of defined standardized scores and the better training of operators, will eliminate as much as possible the diagnostic doubts related to US diagnosis.

The main limitations of our study are represented by the small population sample and by the longitudinal US evaluation in women at risk for accretion. In fact, MRI is generally performed once in pregnancy, so this can increase the US diagnostic accuracy, considering that signs of advanced placental invasion may appear late in pregnancy.

## 5. Conclusions

In conclusion, US performed by experienced operators is a diagnostic tool for PAS. However, large prospective studies are necessary to confirm these results and to develop standardized protocols to improve the outcomes of women affected by PAS disorders.

## Figures and Tables

**Figure 1 diagnostics-12-02769-f001:**
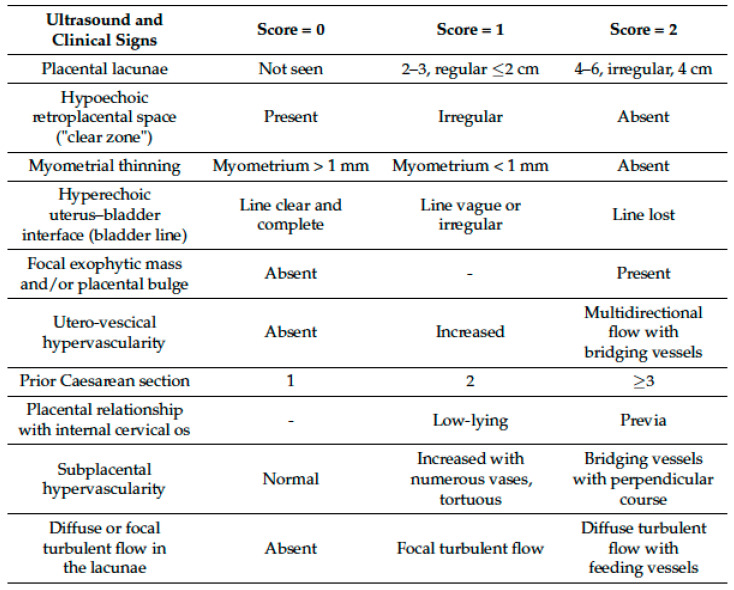
US score for PAS diagnosis.

**Table 1 diagnostics-12-02769-t001:** Distribution of patients’ characteristics according to PAS severity groups.

Characteristics and Risk Factors	No PAS	Accreta	Increta	Percreta	*p*-Value
Number of patients (%)	79 (75)	14 (13)	7 (6.7%)	4 (3.8%)	
Mean age (SD) ^1^	37 (5.5)	38 (5.6)	40 (7.5)	38 (8)	0.73
No previous cesarean (%)	48 (60.8)	2 (14.3)	1 (14.3)	0	<0.001
1 previous cesarean (%)	24 (30.4)	7 (50)	2 (28.6)	0	<0.001
2 previous cesareans (%)	5 (6.3)	3 (21.4)	1 (14.3)	3 (75)	<0.001
>2 previous cesareans (%)	2 (2.5)	2 (14.3)	3 (42.8)	1 (25)	<0.001
Major placenta previa (%)	45 (60)	11 (78.6)	7 (100)	4 (100)	0.02
Minor placenta previa (%)	34 (43)	3 (21.4)	0	0	0.02
Previous courettage (%)	25 (31.6)	4 (28.6)	1 (14.3)	2 (50)	0.64
No previous courettage (%)	54 (68.4)	10 (71.4)	6 (85.7)	2 (50)	0.64

^1^ SD: standard deviation.

**Table 2 diagnostics-12-02769-t002:** Results of the logistics model.

Variable	Coefficient	SD ^1^	*p* Value
Intercept	−3.95	0.87	<0.001
Previous CS ^2^	1.20	0.33	<0.001
Previous courettage	−0.08	0.041	0.83
Previous myomectomy	2.87	1.38	0.03
Major placenta previa	1.9	0.82	0.02

^1^ SD: standard deviation; ^2^ CS: cesarean section.

**Table 3 diagnostics-12-02769-t003:** Histological/surgical diagnosis vs. US/MRI in differentiating PAS from NO-PAS cases.

		US Diagnosis		MRI Diagnosis	
Histological/ Surgical diagnosis		No PAS	PAS	No PAS	PAS
	NO PAS (%)	70 (88.6)	9 (11.4)	70 (88.6)	9 (11.4)
	PAS (%)	2 (8)	23 (92)	3 (12)	22 (88)

**Table 4 diagnostics-12-02769-t004:** Accuracy, sensitivity, specificity, and positive and negative predictive values for US and MRI.

	US	MRI
Accuracy	0.89	0.88
Sensitivity	0.92	0.88
Specificity	0.89	0.89
PPV ^1^	0.72	0.71
NPV ^2^	0.97	0.95

^1^ PPV: positive predictive value; ^2^ NPV: negative predictive value.

**Table 5 diagnostics-12-02769-t005:** Diagnostic ability of US and MRI in correctly identifying the degree of PAS.

		US Diagnosis				MRI Diagnosis			
Histological/ Surgical diagnosis		No PAS	Accreta	Increta	Percreta	No PAS	Accreta	Increta	Percreta
	No PAS (%)	70 (88.6)	9 (11.4)	0	0	70 (88.6)	9 (11.4)	0	0
	Accreta (%)	2 (14.3)	12 (85.7)	0	0	2 (14.3)	10 (71.4)	2 (14.3)	0
	Increta (%)	0	3 (42.8)	2 (28.6)	2 (28.6)	1 (14.3)	4 (57.1)	0	2 (28.6)
	Percreta (%)	0	0	0	4 (100)	0	1 (25)	0	3 (75)

**Table 6 diagnostics-12-02769-t006:** US and MRI: PAS/no-PAS distinction according to placental location.

		Anterior	Posterior	Total
US: PAS/no-PAS distinction	Error (%)	5 (5.8)	6 (33.3)	11 (10.6)
	Correct (%)	81 (92.2)	12 (66.7)	93 (89.4)
	Total	86	18	104
		Accuracy for anterior placenta	Accuracy for posterior placenta	Difference significance (chi-squared test)
		0.94	0.67	*p* = 0.0024
MRI: PAS/no-PAS distinction	Error (%)	6 (7)	6 (33.3)	12 (11.5)
	Correct (%)	80 (93)	12 (66.7)	92 (88.5)
	Total	86	18	104
		Accuracy for anterior placenta	Accuracy for posterior placenta	Difference significance (chi-squared test)
		0.93	0.67	*p* = 0.0055

**Table 7 diagnostics-12-02769-t007:** US and MRI: PAS degree’s distinction according to the placental location.

		Anterior	Posterior	Total
US	Error (%)	9 (10.5)	7 (38.9)	16 (15.4)
	Correct (%)	77 (89.5)	11 (61.1)	88 (84.6)
	Total	86	18	104
		Accuracy for anterior placenta	Accuracy for posterior placenta	Difference significance (chi-squared test)
		0.89	0.61	*p* = 0.0074
MRI	Error (%)	13 (15.1)	8 (44.4)	21 (20.2)
	Correct (%)	73 (84.9)	10 (55.6)	83 (79.8)
	Total	86	18	104
		Accuracy for anterior placenta	Accuracy for posterior placenta	Difference significance (chi-squared test)
		0.85	0.55	*p* = 0.0126

## Data Availability

The data presented in this study are available on request from the corresponding authors.

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
