# Peer review of "Placenta Accreta Spectrum (PAS) Disorder: Ultrasound versus Magnetic Resonance Imaging"

_diagnostics, 2022, doi:10.3390/diagnostics12112769_

Round 1
Reviewer 1 Report
The authors evaluated the diagnostic accuracy of US and MRI in the diagnosis of PAS and to define the most relevant characteristics that can predict placental invasion. They concluded that US, when properly implemented with the application of defined and standardized scores, can be superior to MRI and absolutely sufficient for the diagnosis of PAS.
The paper is good and well written. The data supports the conclusions, and the message is simple and well described. I have suggested a few minor changes for improvement.
Line 71 - Is there anything missing after "low"? what is low?
Line 125-126 - The statement does not seem to match what is in the table with respect to previous myomectomy.
line 154-155 - It is unclear how those numbers were generated.
Table 2 - the superscript 2 with CS is not in a footnote
Should leave a space between lines 131 and 132, as well as between lines 143 and 144.
The discussion and conclusion have no statements on "to define the most relevant characteristics that can predict placental invasion" which is mentioned in the abstract.
Reviewer 2 Report
Dear Authors,
The presented study tackles an issue of Placenta Accreta Spectrum (PAS) Disorder. The validity of Ultrasound versus Magnetic Resonance Imaging is still unclear. I have read the article with a great interest. The study was conducted reliably with appropriate selection of tests. Overall, I think that this article should be published, however some issues require complementary information:
1. Abstract- Results section should include some more information like statistical significance of findings.
2. The Introduction is insufficient. You should state prevalence of PAS, what you can find in ultrasound and MRI imaging and why (the information stated in “Figure 1”).
3. Verse 75-78- I suggest including the information about the technique of ultrasound. Was it transabdominal, transvaginal, both? What kind of doppler was used?
4. From verse 91- It’s not a Figure 1 but Table 1
5. Verse 109 and 110- The statement about guidelines should be at the beginning – not in “Definitive Diagnosis”
6. I suggest including the information about the methods of statistical analysis in Materials and Methods section.
7. I suggest including the information about medical experience of the doctors (years of practice).
8. I suggest including some information about the timing of diagnosis (weeks of gestation)
9. Table 8. - I suggest deleting that table- It’s not a review study. Just state the results in Discussion.
10. I suggest including at the beginning of Discussion major findings from your study.
